# Real-world wintertime CO, N$_2$O and CO$_2$ emissions of a Central European village

László Haszpra[1,2], Zoltán Barcza[3,4,5], Zita Ferenczi[6], Roland Hollós[3,7,8], Anikó Kern[5,9], Natascha Kljun[10]

[1]Institute for Nuclear Research, Debrecen, 4026, Hungary
[2]Institute of Earth Physics and Space Science, Sopron, 9400, Hungary
[3]Department of Meteorology, Institute of Geography and Earth Sciences, ELTE Eötvös Loránd University, Budapest, 1117, Hungary
[4]Excellence Center, Faculty of Science, ELTE Eötvös Loránd University, Martonvásár, 2462, Hungary
[5]Faculty of Forestry and Wood Sciences, Czech University of Life Sciences Prague, Prague, 165 21, Czech Republic
[6]Hungarian Meteorological Service, Budapest, 1024, Hungary
[7]Agricultural Institute, Centre for Agricultural Research, Martonvásár 2462, Hungary
[8]Doctoral School of Environmental Sciences, ELTE Eötvös Loránd University, Budapest 1117, Hungary
[9]Department of Geophysics and Space Sciences, Institute of Geography and Earth Sciences, ELTE Eötvös Loránd University, Budapest, 1117, Hungary
[10]Centre for Environmental and Climate Science, Lund University, Lund, 223 62, Sweden

*Correspondence to:* László Haszpra (haszpra.l@gmail.com)

**Abstract.** Although small rural settlements are only minor individual sources of greenhouse gases and air pollution, their high overall occurrence can significantly contribute to the total emissions of a region or country. Emissions from a rural lifestyle may be remarkably different than that of urban and industrialized regions, but nevertheless they have been hardly studied so far. Here, flux measurements at a tall-tower eddy covariance monitoring site and the footprint model FFP are used to determine the real-world wintertime CO, N$_2$O, and CO$_2$ emissions of a small village in western Hungary. The recorded emission densities, dominantly resulting from residential heating, are 3.5 µg m$^{-2}$ s$^{-1}$, 0.043 µg m$^{-2}$ s$^{-1}$, and 72 µg m$^{-2}$ s$^{-1}$ for CO, N$_2$O, and CO$_2$, respectively. While the measured CO and CO$_2$ emissions are comparable to those calculated using the assumed energy consumption and applying the according emission factors, the nitrous oxide emissions exceed the expected value by a magnitude. This may indicate that the nitrous oxide emissions are significantly underestimated in the emission inventories, and modifications in the methodology of emission calculations are necessary. Using a 3-dimensional forward transport model, we further show that, in contrast to the flux measurements, the concentration measurements at the regional background monitoring site are only insignificantly influenced by the emissions of the nearby village.

## 1 Introduction

Climate change, primarily caused by the accumulation of greenhouse gases (GHG) in the atmosphere, is one of the biggest challenges humanity faces. In addition to the direct meteorological consequences, it also manifests in different economical and societal problems including food and water insecurity, migration, political crises, loss of biodiversity, etc. (IPCC, 2014). For the development of an effective climate change mitigation strategy, we need to know the amount of greenhouse gases emitted by each source. At country level, the emission is

calculated based on statistical activity data and emission factors suggested by international guidelines (IPCC, 2019). However, emission inventory guidelines cannot specify emission factors for each activity and specific conditions/circumstances, resulting in distortion and uncertainty of the officially reported inventory values, the essential input of the European Union's emission control policy. The climatic consequences of the anthropogenic GHG emission depend on the actual amount of GHG emissions rather than the emissions calculated based on

potentially uncertain emission factors, therefore it is highly desirable to validate and improve the accuracy of emission factors. Industrial emissions can be estimated with relatively low uncertainty. Household emissions, however, may vary largely depending on the available infrastructure, socio-economic conditions, and cultural traditions, especially for small settlements. Although (mega)cities dominate the anthropogenic greenhouse gas emission (Moran et al., 2018), the large number of small settlements with poorly constrained emissions and the

scarce direct measurements at village-environment-scale (Fachinger et al., 2021) contribute to the uncertainty of the estimates of the total anthropogenic emission.

Since 1993, the tall-tower greenhouse gas monitoring station Hegyhátsál has been operated for the Global Atmosphere Watch program of the World Meteorological Organization and the global cooperative air sampling network of the National Oceanic and Atmospheric Administration, U.S.A., in a rural environment in Hungary. In

addition to the continuous monitoring of the concentration of direct ($CO_2$, $CH_4$, $N_2O$) and indirect (CO) greenhouse gases, the tower is also equipped with eddy covariance (EC) systems to measure the surface-atmosphere exchange of certain gases. The site is located as far from any major anthropogenic sources as possible in the densely populated Central Europe. However, from time to time, the footprint area of the EC systems partly overlaps with the area of a nearby small village (Barcza et al., 2009). There is no industrial or

notable commercial activity in the village; therefore, it is an ideal place to estimate one of the most uncertain terms of the emission inventories of small settlements: residential heating. EC measurement-based emission mapping is common in urban environments (see e.g. Rana et al. (2021), and references herein). In this study, we show that long-term data series at a tall tower can be used for the determination of the emissions of a small settlement occupying only a minor area of the footprint area of the EC system. For the derivation of the

residential emissions, the hourly nitrous oxide ($N_2O$), carbon monoxide (CO), and carbon dioxide ($CO_2$) fluxes measured at the tower during the winter seasons (Dec-Feb) between December 2015 and February 2021 were used. To the best of our knowledge, we are the first to attempt to apply this technique in a rural, natural environment for the determination of the emission of a village covering only a small part of the footprint area of the measurements.

As footprints of the eddy covariance and concentration measurements differ by magnitudes (Gloor et al., 2001; Kljun et al., 2002; Vesala et al., 2008; Barcza et al., 2009), the question arises of whether and to what extent the emissions from the nearby village impact the concentration measurements at the monitoring station. A forward transport model was deployed to answer this question.

**2 Methods and measurements**

**2.1 Basic concept**

The eddy covariance (EC) technique is widely used for the determination of surface-atmosphere flux of atmospheric components within the footprint of the measurements (Franz et al., 2018; Papale, 2020). Although

the majority of EC systems are used for the monitoring of gas exchange of different ecological systems, there is a growing number of EC sites used for the estimation of urban anthropogenic emissions (see e.g. Grimmond et al., 2002; Vogt et al., 2006; Stagakis et al., 2019; Rana et al., 2021). Usually, the emission density in a city is not spatially homogeneous. An appropriate footprint model can help to attribute the measured flux to the emission in specific source areas.

The flux footprint area depends on the measurement height of the EC system and the actual meteorological and surface conditions. Flux footprints of EC measurements performed on a tower of almost 100 m tall may cover an area of up to a hundred square kilometers (Barcza et al., 2009; Desai et al., 2015; Satar et al., 2016; Chi et al., 2019). Within this area, the village in focus may cover only a few percentages. Our concept assumes that, during the winter season, both the "natural" landscape (vegetated area, i.e. agricultural fields, forests) and the built-up area (i.e. the village) are homogeneous from the point of view of emission density. The measured flux ($F_{measured}$) is then the combination of the fluxes originating from the built-up areas ($F_{village}$) including the houses, farm buildings, backyards, roads and parks within the village, and those of the non-residential areas, which we denote here as "natural" landscape ($F_{natural}$):

$$F_{measured} = \alpha * F_{village} + (1 - \alpha) * F_{natural} , \qquad (1)$$

where $\alpha$ is the contribution of the village within the footprint area. Note that the contribution of the surface flux to the measured flux is not uniform within the footprint area (Schmid, 1994; Kljun et al., 2002; Vesala et al., 2008). The weighted contribution of a surface source at a specific unit area to the measurement at the tower at each time step can be estimated by using the footprint function value at the given point. The integral of the footprint function over the infinite x-y plane equals one. Hence, with a suitable footprint model, if $F_{natural}$ is known, then $F_{village}$, the emission density of the village within the footprint area can be calculated as follows:

$$F_{village} = (F_{measured} - (1 - \alpha) * F_{natural}) / \alpha \qquad (2)$$

In a next step, the influence of the emission of the village on the concentration measurements at the monitoring site can be calculated using an appropriate transport model. The realization of this approach requires surface-atmosphere flux measurements, land cover information, as well as footprint and transport models.

**2.2 Monitoring site and instrumentation**

The EC measurements used in this study are carried out on a 117 m tall, free-standing TV/radio transmitter tower owned by Antenna Hungária Corp. The tower is located in a fairly flat region of western Hungary, close to the western edge of the Pannonian Basin (46°57′N, 16°39′E, 248 m asl), in the vicinity of the small village called Hegyhátsál being in the focus of the present study (Fig. 1).

The eddy covariance system is mounted on the tower at 82 m above the ground, on an instrument arm of 4.4 m length projecting to the north. The disturbance of the flow pattern during southerly winds is corrected as described in Barcza et al. (2009). The eddy covariance system has been monitoring the vertical flux of $CO_2$ since 1997, and that of $N_2O$ and CO since 2015. The EC system is based on a GILL R3-50 research ultrasonic anemometer (GILL Instruments Ltd, Lymington, U.K.), a Model 913-0014 Enhanced Performance fast-response

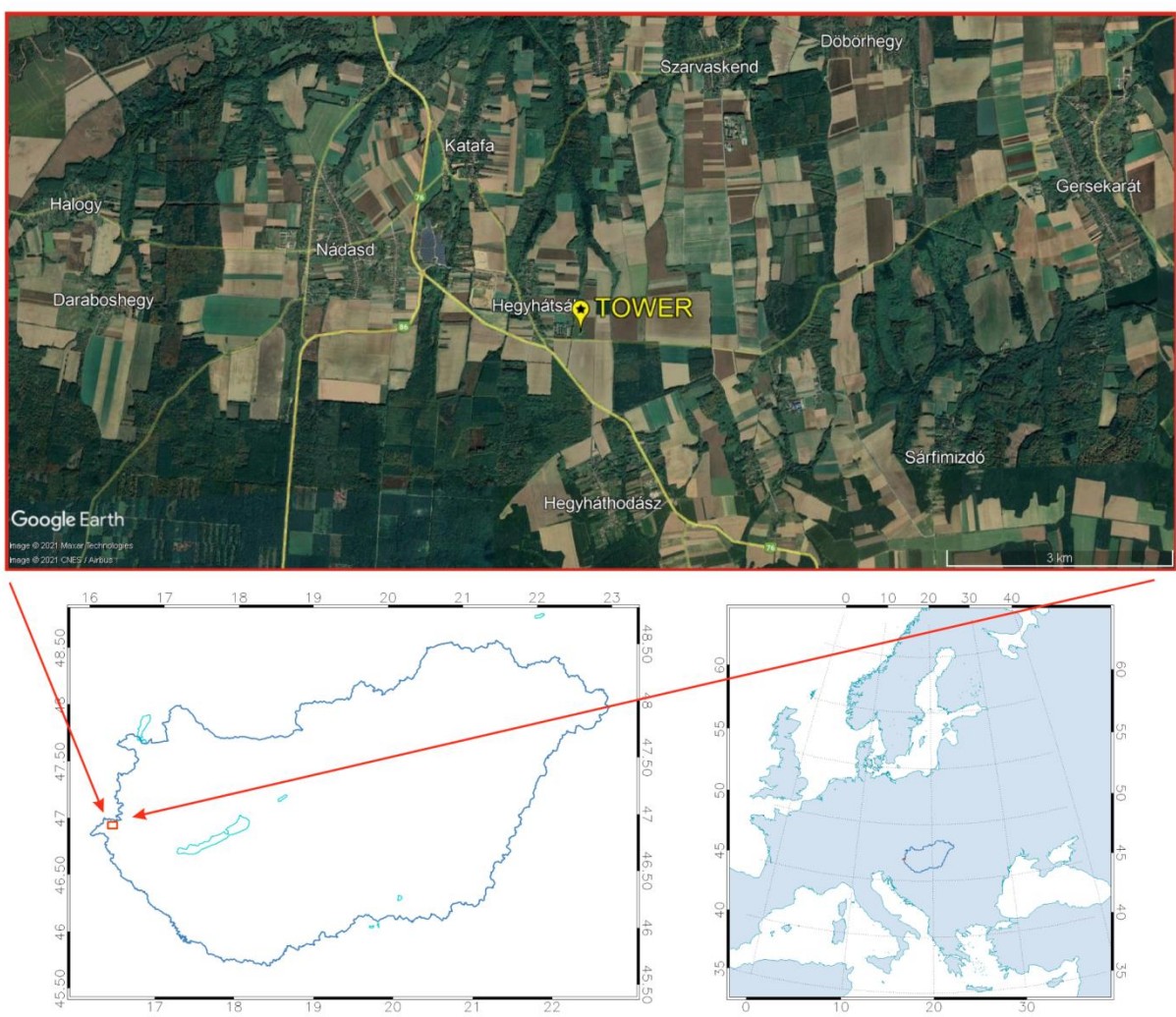

**Figure 1: Location and the surrounding region of Hegyhátsál tall-tower GHG monitoring site on a Google Earth satellite image.**

N$_2$O/CO/H$_2$O analyzer (Los Gatos Research Ltd., San Jose, CA, U.S.A.) with fast-flow optional accessories, and a Model LI-6262 fast response infrared CO$_2$/H$_2$O analyzer (Li-Cor Inc., Lincoln, Nebraska, USA).

In addition to the eddy covariance measurements, the concentration of carbon dioxide and the basic meteorological parameters (wind speed and direction, air temperature, relative humidity) are continuously measured at four elevations along the tower (10 m, 48 m, 82 m, 115 m). In this study, we use the term "concentration" as a synonym of the actually measured "dry mole fraction". For a detailed description of the site and instrumentation see Haszpra et al. (2001; 2018) and Barcza et al. (2020).

### 2.3 Surface-atmosphere flux calculation

The measurements performed at the tower allow the calculation of the turbulence parameters (vertical wind speed and concentration fluctuations) necessary for the calculation of the vertical fluxes of the substances studied. A detailed description of the methodology for these calculations can be found in Haszpra et al. (2001), and Barcza et al. (2020). The EC system and the data evaluation software provide averaged hourly flux values.

A disadvantage of tall-tower EC systems is that they may be decoupled from the surface by low-level inversions

from time to time (Desai et al., 2015; Chi et al., 2019). At our monitoring station, such conditions are found especially during winter. In these situations, the EC systems cannot provide the actual surface-atmosphere flux data. To avoid decoupled measurements in cases of low-level inversion, information on the height of the boundary layer was derived from the ERA5 reanalysis dataset of the European Centre for Medium-Range Weather Forecasts (Copernicus Climate Change Service, 2017) for the grid-point nearest to the monitoring site

with hourly resolution. Taking into account the elevation of the EC system of 82 m above the ground, we removed all flux values from the quality-checked data series (Haszpra et al., 2005; 2018; Barcza et al., 2020) when the top of the boundary layer was below 100 m. Removing these data from the data series, 6371 hourly data points (49.0 % of the total 13008 winter hours) remained for the study for the six winter seasons (2015/2016-2020/2021) evaluated in this study.

**2.4 Environmental and land cover information**

The project aims at the determination of the wintertime, residential heating dominated emission of Hegyhátsál village. The village is located in the west-to-northwest sector, in 400-1200 m distance from the tower. It has 151 inhabitants in 89 households (Hungarian Central Statistical Office, 2019). There is no industrial or notable commercial activity in the village. Approximately half of the single-family houses of the village are connected to

the natural gas distribution network and use this fuel also for heating purposes. The other half of the households use solid fuels for heating. Taking into account the socioeconomic conditions in the region, it is reasonable to assume that heating appliances for biomass or other solid fuels are available and occasionally used even in the households connected to the natural gas network.

The land cover of the region of the monitoring tower consists of a mixture of agricultural fields and small forest

patches. In addition to Hegyhátsál, the other neighboring villages are about 3 km away from the tower to the north (Katafa), northwest (Nádasd), and south (Hegyháthodász). The nearest settlement worth mentioning in the eastern sector is Gersekarát, located more than 7 km from the tower (Fig. 1). There is hardly any commercial or industrial activity in this dominantly agricultural region.

The local roads connecting the small settlements carry only little traffic (300-600 vehicle units per day). The

only major road in the region is the 2x1 lane trans-European E65 running northwest-southeast with 4700 vehicle units per day (Magyar Közút, 2019). Its closest point to the monitoring site is about 500 m to the southwest (Fig. 1).

The prevailing wind directions in winter are northeasterly and southwesterly (Fig. 2), although the monitoring station is located in the zone of westerly wind patterns. However, the Alps rising approximately 100 km to the

west of the station significantly modify the regional wind pattern.

For the identification of the land cover type of the potential source areas of the surface-atmosphere fluxes, the National Ecosystem Base Map of Hungary (NÖSZTÉP) (Tanács et al., 2019) was used. This dataset has 56 categories at Level-3 with a spatial resolution of 20 m x 20 m. Within the area of our interest three Level 1 NÖSZTÉP categories occur: urban, cropland, and forest. In our study, cropland and forested land cover types are

considered as "natural" landscape areas, while the areas labeled as urban (buildings, roads and other artificial surfaces, vegetated areas in an artificial environment [e.g. backyards, parks, etc.]) represent the villages. There

are 1645 grid cells covering the area of Hegyhátsál village, which corresponds to its area of 65.8 hectares (Fig. 3).

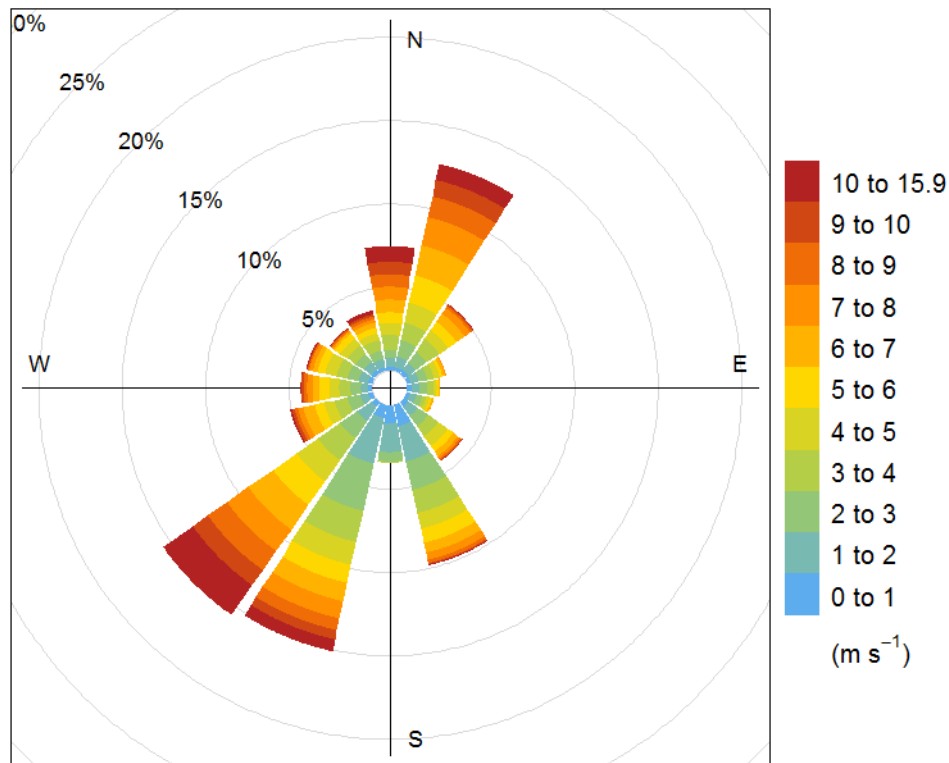

**Figure 2: Wintertime (Dec-Feb) frequency distribution of wind directions at 82 m height at the Hegyhátsál tall-tower monitoring site between December 2015 and February 2021. The village is located to the northwest of the tower.**

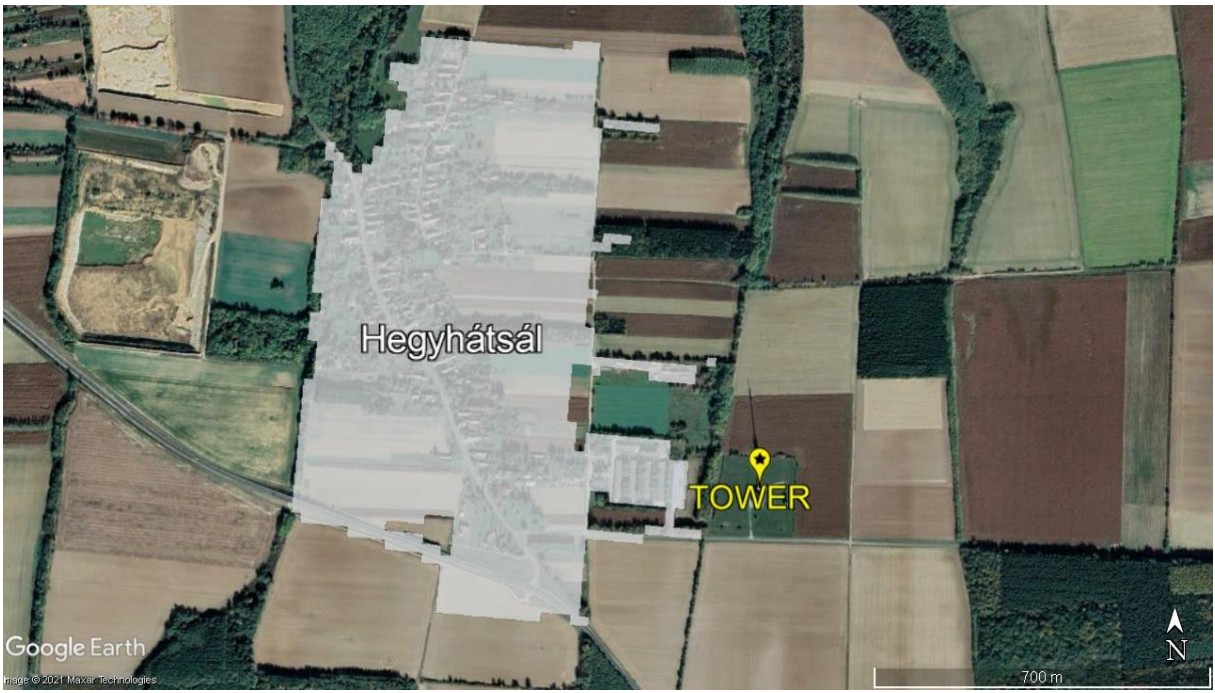

**Figure 3: Definition of the territory of the village based on land cover information. The shaded area covers the buildings, roads in the village, and vegetated areas in artificial environment (parks, backyards, etc.).**

**2.5 Footprint model**

The flux footprint function is a probability density function, describing the relative contribution from each element of the (mainly) upwind surface area source to the measured flux. For the calculation of the source area (footprint) of the flux measurements, the 2-dimensional Flux Footprint Prediction (FFP) model of Kljun et al. (2015) has been applied. The model is based on the LPDM-B backward Lagrangian stochastic particle dispersion model valid for a wide range of atmospheric conditions (Kljun et al., 2002; 2004a; 2004b). While FFP is significantly less resource-intensive than LPDM-B, it is still applicable for stable, neutral, and convective conditions. The model performance was amongst the best in a test of several footprint models against data from a tracer release experiment (Heidbach et al., 2017).

The input parameters of the model are the measurement height above displacement height ($z_m$), the roughness length ($z_0$), the Obukhov length (L), the standard deviation of the lateral wind speed ($\sigma_v$), the friction velocity ($u_*$), and the height of the boundary layer (h). The wind direction is an optional input parameter but it is needed for the geographical localization of the source areas. Displacement height was considered to be negligible due to the lack of vegetation in wintertime (see van der Kwast et al., 2009), thus the observation height was used to approximate $z_m$. The Obukhov length, the standard deviation of the lateral wind speed, the friction velocity, and the wind direction are directly measured or can be calculated from the measurements. The boundary layer height is available from the ERA5 reanalysis data set (see above) for the region of the tower. The roughness length is assumed to equal 0.15 m based on an earlier study (Barcza et al., 2009).

FFP assumes the stationarity and horizontal homogeneity of the flow over the time periods of the flux calculations (one hour in our case) and does not include roughness sublayer dispersion near the ground (negligible in this case) nor dispersion within the entrainment layer at the top of the convective boundary layer. The scaling parametrization also sets some limitations. In this study, the model was used with the following restrictions:

$$20 z_0 < z_m < h_e \tag{3}$$
$$-15.5 \leq z_m/L \tag{4}$$
$$u_* \geq 0.2 \text{ m s}^{-1}, \tag{5}$$

where $h_e$ is the height of the entrainment layer. Accepting that typically $h_e \approx 0.8h$ (Holtslag and Nieuwstadt, 1986; Kljun et al., 2015), (3) does not reduce the available flux data as the fluxes measured during h < 100 m have already been excluded from the data set (see above). However, (4) and (5) disqualify approximately a third of our measurement data. So, for footprint calculation, 4277 hourly flux data, 32.9 % of the total winter hours, were available. Theoretically, it is possible to set a lower $u_*$ threshold for the FFP model. However, at low $u_*$ the EC systems mounted high above the ground cannot provide the actual surface-atmosphere flux data alone. In such cases, the storage term has to be considered (Haszpra et al., 2005). As the storage term adds considerable uncertainty to the calculated flux due to the noisy signal, it is preferable to avoid low $u_*$ conditions in the tall-tower flux derivation.

The discretized footprint function, i.e. the output of the model at the 90 % footprint contribution level, was integrated for each grid cell of the land cover map giving the contribution of that specific grid cell to the total flux measured at the monitoring site. The footprint function was also integrated over the area of the village ($\alpha$) to

indicate the total contribution of the emission from the village to the measured flux at the monitoring site. To derive α, the integral concerns all grid cells appearing in Fig. 3.

## 2.6 Transport model

For the estimation of the influence of the emission from the village on the concentration measurements at the
monitoring tower, the Graz Lagrangian Model (GRAL v14.8 – Oettl et al., 2002; Oettl, 2015a; Oettl, 2015b; Romanov et al., 2020) has been used. This 3D particle dispersion model was originally developed for the dispersion of pollutants from a road tunnel portal but is suitable to describe the 3-dimensional concentration distribution of area sources. Its input data are wind speed, wind direction, Pasquill-Gifford stability class (determined from the local meteorological measurements), location of the source area relative to the receptor
point, and the yield of the source homogeneously distributed over the area. The simulation was run at 10 m horizontal and 3 m vertical resolution for the period of December 2017 – February 2018 at 1 h temporal resolution.

## 3 Results and discussion

### 3.1 Emissions of the "natural" landscape

The natural sources of carbon monoxide comprise biomass burning, atmospheric oxidation of hydrocarbons, and direct biogenic emissions (Zheng et al., 2019). Open biomass burning (e.g. stubble burning) is prohibited in the study region. Atmospheric oxidation of hydrocarbons requires hydroxyl radicals. Hydroxyl radicals form in photochemical processes, therefore their concentration is low in the darkest season of the year. The vegetation is dormant in winter, and biogenic emissions also depend on sufficient light (Bruhn et al., 2013). Hence, it can be
concluded that natural sources of CO are negligible during winter time.

The major natural sources of nitrous oxide are denitrification and nitrification processes in soil and water (Tian et al., 2020). These biochemical processes slow down with decreasing temperature (Benoit et al., 2015; Butterbach-Bahl et al., 2013). Nitrogen addition to agricultural soil enhances nitrous oxide emission, which is relevant in our case as the surrounding region of the tower is dominated by croplands (Barcza et al., 2009).
Although the average temperature of +1.6 °C during the study period is rather low for biochemical activities, the croplands (i.e. "natural" landscape) may emit a detectable amount of $N_2O$.

The net ecosystem exchange of carbon dioxide in the winter season is positive in our region (Haszpra et al., 2005; Barcza et al., 2020), i.e. the landscape is a net source. The dominantly dormant vegetation assimilates only a low amount of carbon dioxide. This process might be temperature-dependent during the winter season.
Respiration decreases with decreasing temperature. The result of the two opposing processes, photosynthetic assimilation and respiration, creates the net emission in winter, on average.

For the determination of the emission density of the non-residential landscape (agricultural fields, forests), we selected those footprints where the integrals of the footprint function values over the area of the village (α) were negligible, that is, the emission in the village did not influence the fluxes measured at the monitoring site (at the
90 % footprint contribution level). Obviously, all these 1147 footprints cover areas in the easterly to southerly sector, opposite the village Hegyhátsál. To avoid any contamination from remote settlements, data with a footprint peak location farther than 5000 m were excluded from the evaluation, which left us with 1120

footprints and flux data points. Nevertheless, our selection cannot completely exclude any anthropogenic emissions. The local roads with little traffic and small settlements of a few households still contribute to the

emission of the "natural" landscape.

The frequency distribution functions of all measured fluxes are skewed towards positive fluxes with a few extremely high values (Fig. 4). Therefore, instead of the arithmetic average, the emission density of the "natural landscape" is characterized by the median of the data sets. The medians are not sensitive to extreme outliers, hence we did not apply any arbitrary outlier filtering algorithm.

Our method assumes homogeneous and isotropic spatial distribution of GHG fluxes from the "natural" landscape. To evaluate whether this condition was met, the measured flux values were grouped into wind sectors of 22.5-degrees, and the median value was calculated for each sector (see Fig. S1 in Supplementary material). The directional medians were compared with the overall median of the data set using the asymptotic K-sample Brown-Mood median test. Only the median of the southwest direction deviated significantly ($p<0.05$) from the

overall median, possibly due to a flux signal from the major road in that direction. Discrimination of a specific direction would have been inconsistent with the otherwise applied α-based filtering, therefore we calculated how much the somewhat higher fluxes from this direction contributed to the overall median. For $N_2O$ it was 2 %, while for CO and $CO_2$ it was 8 %. These small deviations hardly influence the calculation of the emission from the village. Therefore, for keeping the consistency of the filtering method, all available data were used for the

calculation of the "natural" landscape. The sector containing the village could not be tested because of the contribution of the village itself. We hence assumed that the overall median of the "natural" GHG fluxes was also applicable for this sector.

The emissions of settlements are reported for the average environmental conditions, therefore, the temperature dependence of the natural GHG emissions was not analyzed in depth in this study. For information, Fig. S2

depicts these temperature dependences. Further, due to the diurnal variation in the meteorological conditions, the number of available data was slightly daytime-biased (Fig. S3). Considering the uncertainty of the data, we do not apply any bias correction in this study.

With the above assumptions, the following emission densities were obtained for CO, $N_2O$, and $CO_2$ for the natural landscape: 139 ng $m^{-2}$ $s^{-1}$, 5.9 ng $m^{-2}$ $s^{-1}$, and 12 μg $m^{-2}$ $s^{-1}$, respectively. Table 1 also lists the lower and

upper quartiles of the emission densities, to provide an impression of the uncertainty of the median values.

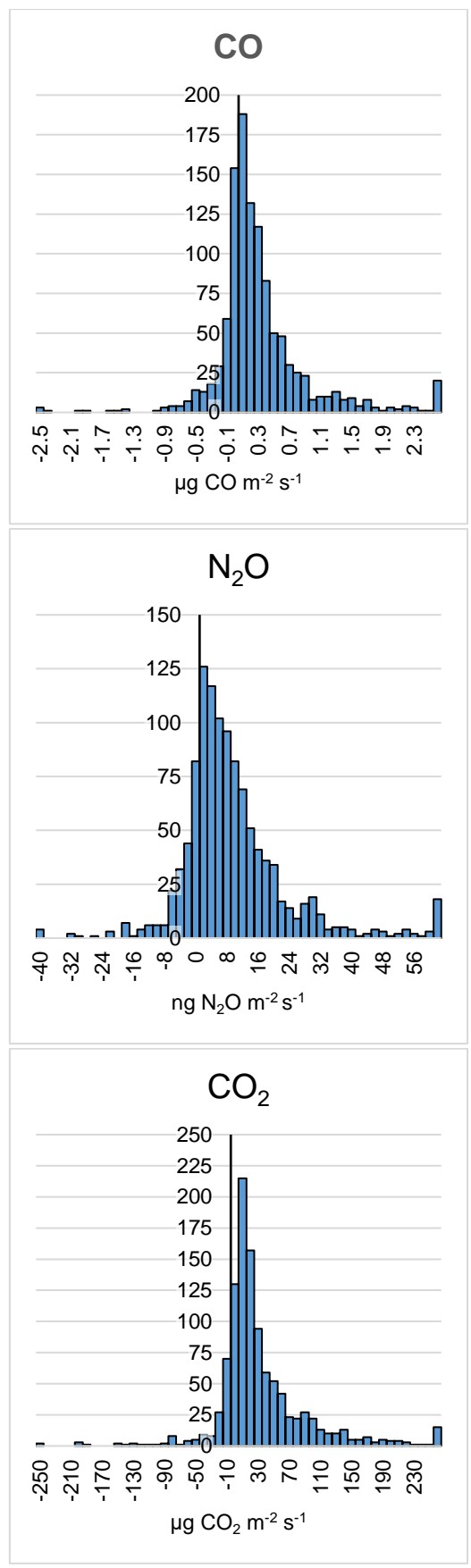


**Figure 4: Frequency distributions of wintertime "natural" landscape (agricultural fields, forests) emissions.**

### 3.2 Emission from the village

For the determination of the emission density of the village using the top-down approach (i.e. estimation of the emission through atmospheric measurements), we should ideally select flux data points when the footprint of the flux measurements exactly covers the area of the village, not missing any part of it and not including anything but the village itself. Due to the location, size, and shape of the village, this was, however, not possible, as all footprints also included non-village contributions. Increasing the required minimum contribution of the village (the integral of the footprint function over the area of the village, α in Eq. (1) and Eq. (2)) results in a decrease in the number of the available hourly flux data points (Fig. 5). The emission density of the village has been calculated for ≥25 %, and ≥30 % footprint weighted coverage (α≥0.25 and α≥0.30) using the constant $F_{natural}$ and the actual hourly α values in Eq. (2). Fig. 6 shows an example of the footprint function with large α value. In addition to the median flux, Table 1 lists the estimated lower and upper quartiles, and the number of footprints available for the calculations. The low number of cases (64 and 44, respectively) is also due to the prevailing wind directions that avoid the village (see Fig. 2). Nevertheless, the small difference between the emission densities derived for ≥25 % and ≥30 % footprint weighted coverages indicates that these contributions can be used to satisfactorily derive the emission densities of the village. At ≥30 % footprint weighted coverage, the interquartile ranges are a bit narrower, suggesting slightly lower uncertainty. Fig. S4 in the Supplementary material shows how $F_{village}$ converges as $\alpha_{min}$ increases.

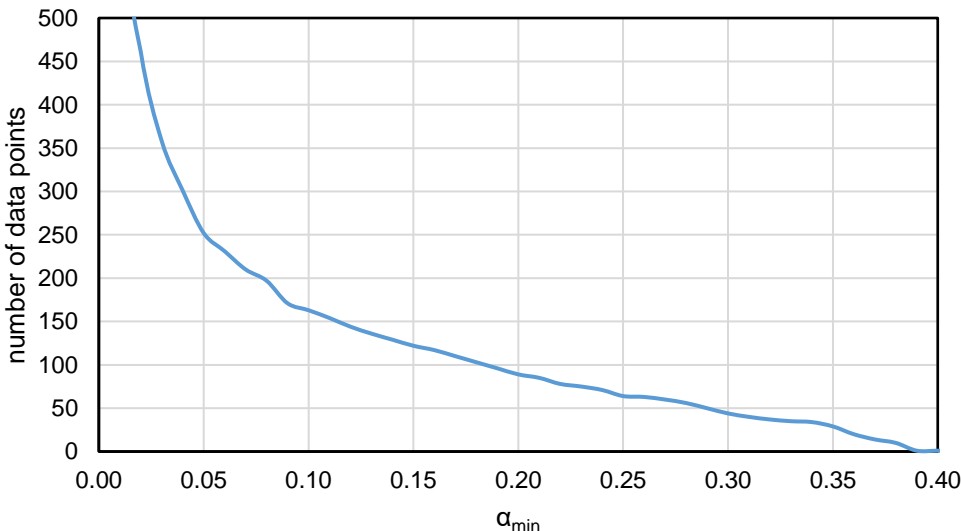

**Figure 5: The number of hourly data points when the integral of the footprint function over the village area (α) was greater than or equal to the value indicated in the x-axis. At x=0, the total number of data points is included (4277 – outside axis range).**

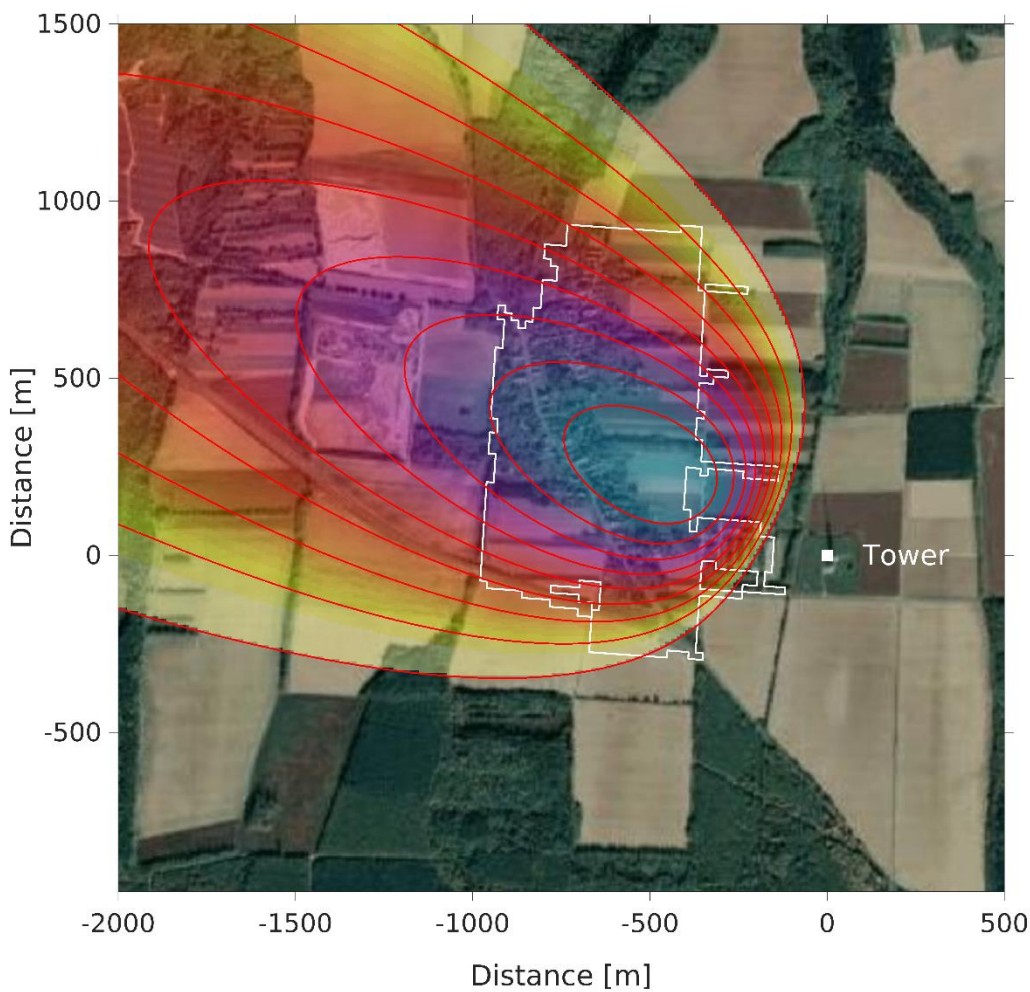

310

**Figure 6: Example of a single flux footprint (10:00-11:00 h LST, 27 December 2016). The footprint contour lines (red) show 10-90% contribution to the flux measurement, in 10% steps. The integral of the footprint function over the village (α, outlined in white) is 0.282 for this case. (Map source: Google Earth.)**

315

The few data available for the calculation of $F_{village}$ are concentrated almost exclusively in the daytime period of 9-15 h local standard time (57 of 64 and 42 of 44, respectively), which may distort the calculated emissions. Although there are no quantitative data on the diurnal heating/cooking behavior of the populations, it can be 320 assumed that our data avoid both the nighttime low emission periods and the morning/evening high emission periods, and hence should be representative for the diurnal average emission values. This hypothesis cannot be tested with the available data, and the resulted uncertainty must be taken into account in the interpretation of the results.

The correlation between the air temperature and the top-down emission density of CO, $N_2O$, and $CO_2$ is negative 325 and statistically significant at $p<0.05$ significance level (-0.369, -0.343, and -0.480, respectively). The negative correlation between the air temperature and the emission supports that the measured flux originates dominantly from residential heating, which is more intensive at low temperatures. This assumption is further supported by

the high positive linear correlation (+0.503) between CO and $CO_2$ emissions. As this study provides emissions of the settlement for average environmental winter conditions, we did not discuss the temperature dependence here. For indicative purpose, the temperature dependence of the emission from the village can be found in the Supplementary material (Fig. S5).

Taking into account the area of the village (65.8 ha) and the median emission densities presented for 30 % coverage in Table 1, the total wintertime (3 months) CO, $N_2O$, and $CO_2$ emissions of the village are 17.9 Mg (metric ton), 0.216 Mg, and 364 Mg, respectively (Table 2).

**Table 1. Emission density of the natural landscape (0 % village coverage) and that of the village calculated at ≥25 % and ≥30 % footprint weighted coverage of the village, respectively. n gives the number of footprints available for the calculations. Q25, Q50, and Q75 indicate the lower quartile, the median, and the upper quartile of the emission densities calculated on the basis of the hourly flux values.**

| village contribution (%) number of footprints | | 0 % (n=1120) | ≥25 % (n=64) | ≥30 % (n=44) |
|---|---|---|---|---|
| CO [µg m$^{-2}$ s$^{-1}$] | Q25 | -0.01 | 1.5 | 2.1 |
| | **Q50** | **0.14** | **3.4** | **3.5** |
| | Q75 | 0.40 | 6.0 | 6.4 |
| $N_2O$ [ng m$^{-2}$ s$^{-1}$] | Q25 | 0.9 | 8.6 | 11.6 |
| | **Q50** | **5.9** | **42.7** | **42.7** |
| | Q75 | 13.2 | 82.5 | 69.4 |
| $CO_2$ [µg m$^{-2}$ s$^{-1}$] | Q25 | 0 | 20 | 7 |
| | **Q50** | **12** | **86** | **72** |
| | Q75 | 42 | 282 | 219 |

Alternative emission estimates can be obtained using a bottom-up method, i.e. using published or expert-based emission factors. Most of the buildings in the village are several decades old single-family houses, traditional brick constructions without insulation. According to the expert estimates, such a house of average size may need approximately 57 GJ energy for winter heating (personal communication, Unit of National Emission Inventories, Hungarian Meteorological Service). For the 89 households of the village, this results in approximately 5 TJ during a winter season. As only half of the houses are connected to the natural gas network, a maximum of 2.5 TJ energy may come from natural gas and a minimum of 2.5 TJ originates from solid fuels, respectively. (Liquid fuel is not used for residential heating in Hungary.) The default emission factors for natural gas are 26 kg CO TJ$^{-1}$, 0.1 kg $N_2O$ TJ$^{-1}$, and 56.1 Mg $CO_2$ TJ$^{-1}$, respectively, while for solid fuels they are around 4 Mg CO TJ$^{-1}$, 4 kg $N_2O$ TJ$^{-1}$, and 100 Mg $CO_2$ TJ$^{-1}$, respectively, depending on the actual fuel type (wood, lignite, etc.) (IPCC, 2006; European Environmental Agency, 2019). Assuming these values, the overall heating emissions are estimated as 10 Mg, 10 kg, and 390 Mg for CO, $N_2O$, and $CO_2$, respectively. As it can be assumed that even the households with access to natural gas use some solid fuels for heating, the real emission values may be somewhat higher. In the extreme case, if no natural gas would be used at all, the corresponding values were 20 Mg, 20 kg, and 500 Mg for CO, $N_2O$, and $CO_2$.

Under the given climatic conditions, the heating season starts around mid-October and lasts until mid-April. Based on the heating day distribution, 65-70 % of the heating energy is used during December-February, i.e. the

study period. A conservative estimation of 10-15 % share of solid fuels in the households accessing natural gas would give the approximate emission values of 8 Mg, 8 kg, and 310 Mg for CO, $N_2O$, and $CO_2$, respectively, for the December-February period (Table 2).

**Table 2. Winter season emissions of Hegyhátsál village calculated applying the top-down (present study) and bottom-**
**up approach, and their ratios.**

|  | top-down (TD) estimation | bottom-up (BU) estimation | TD/BU |
|---|---|---|---|
| CO | 18 Mg | 8 Mg | 2.2 |
| $N_2O$ | 216 kg | 8 kg | 27 |
| $CO_2$ | 364 Mg | 310 Mg | 1.2 |

Although these statistics-based, bottom-up numbers seem to underestimate the CO and $CO_2$ emissions calculated by the top-down approach, – when taking into account the rough estimate of the bottom-up emission and the
uncertainties of the top-down approach – the results are similar enough to support the applicability of our method. Our measurements show a higher $CO:CO_2$ emission ratio (0.049 vs. 0.026), which indicates the contribution of incomplete combustion that points to biomass rather than natural gas burning. The measurement-based (top-down) $N_2O$ emission is 27-times higher than the statistical-based (bottom-up) one. Even if we assume the upper limit of 15 kg $TJ^{-1}$ for the $N_2O$ emission factor from solid fuel, the resulting value is still a magnitude
lower than the measured one. This emission factor may also indicate biomass (or even waste) burning. Our previous study based on concentration measurements (Haszpra et al., 2019) also showed a higher $N_2O:CO_2$ ratio compared to the official emission estimates. It suggests that the $N_2O$ emission factor for residential heating may be significantly underestimated for the actual conditions. The illegal burning of solid household, municipal or agricultural waste, which is not rare in villages of poor socioeconomic conditions (Hoffer et al., 2020), may also
modify the $CO:N_2O:CO_2$ emission ratios. To prove or disprove the presence of waste burning, local measurements of characteristic organics in the atmosphere would be needed.

Although the small sample size implies considerable uncertainty, we tried to estimate if the likely temperature dependence and daily temporal variation of natural emission density ($F_{natural}$) influence the calculated $F_{village}$. For this purpose, the natural fluxes were grouped into 3-hour time windows (0–3 h, 3–6 h, 6–9 h,... local standard
time), and within each time window the data were grouped into 3-degree temperature ranges centrally to 0 °C (-10.5 – -7.5 °C, -7.5 – -4.5 °C,-4.5 – -1.5 °C, -1.5 – +1.5 °C,...). The median flux was calculated for each group including at least 10 data. These median fluxes (Table S1) were used as $F_{natural}$ for the recalculation of $F_{village}$ taking into account the actual temperature and time of the day. The calculated emission densities for CO, $N_2O$, and $CO_2$ from the village at $\alpha \geq 0.3$ are 3.6 µg $m^{-2}$ $s^{-1}$, 36.3 ng $m^{-2}$ $s^{-1}$, and 73 µg $m^{-2}$ $s^{-1}$, respectively, practically
the same as using the constant $F_{village}$ (median of the overall dataset). The $N_2O$ emission is lower by approximately 15 % according to this method but it does not question the conclusion, namely, the official $N_2O$ emission may be significantly underestimated.

**3.3 Influence of local emissions on the regional background concentration measurements**

The Hegyhátsál tall tower monitoring station is registered in the WMO GAW program (https://gawsis.meteoswiss.ch/GAWSIS/#/search/station/stationReportDetails/0-20008-0-HUN) and the ICOS network (https://meta.icos-cp.eu/labeling/) as a regional background monitoring site. As such, it should receive as little direct anthropogenic pollution as possible in the densely populated, highly industrialized Europe. As the footprints of the eddy covariance measurements (see e.g. Fig. 6) and those of the concentration measurements

(Fig. 7) differ significantly, it is appropriate to check by how much the concentration measurements are influenced by the nearby village. We applied the GRAL model for the 2017/2018 winter season to estimate the influence of the village's emissions on the concentration measurements. For the calculations, $F_{village}$ presented in Table 1 for $\alpha \geq 0.3$ was assumed, with a homogeneous distribution over the area of the village. For 1798 hours of 2160 hours of the study period (83.2 %), emissions from the village did not reach the measurement sensor at all,

mainly due to the prevailing wind directions. (The prevailing wind directions were northeasterly and southwesterly, while the village is located in the west-northwest sector relative to the measurement tower (see Fig. 2 and 3). In a few cases with winds from west-northwest, i.e. from the village, the pollution could not reach the measurement height at 82 m above the ground due to a shallow boundary layer.

Fig. 8 shows the frequency distribution of the excess concentrations derived from the emissions in the village.

The excess burden given in mass per volume unit is converted into concentration given in dry mol fraction assuming standard pressure (972 hPa at 248+82 m above sea level) and air temperature of +0.6 °C (average over the period of December 2017 - February 2018). Due to its high emissions relative to the background concentration, the carbon monoxide concentration is the most sensitive to local pollution. The excess concentration exceeds 2 nmol mol$^{-1}$ in only 0.74 % of the hours. Without considering a specific hour of the study

period, the maximum excess was 2.9 nmol mol$^{-1}$. For comparison, the recommended network compatibility of CO concentration measurements within the scope of WMO/GAW network is 2 nmol mol$^{-1}$ (WMO, 2020).

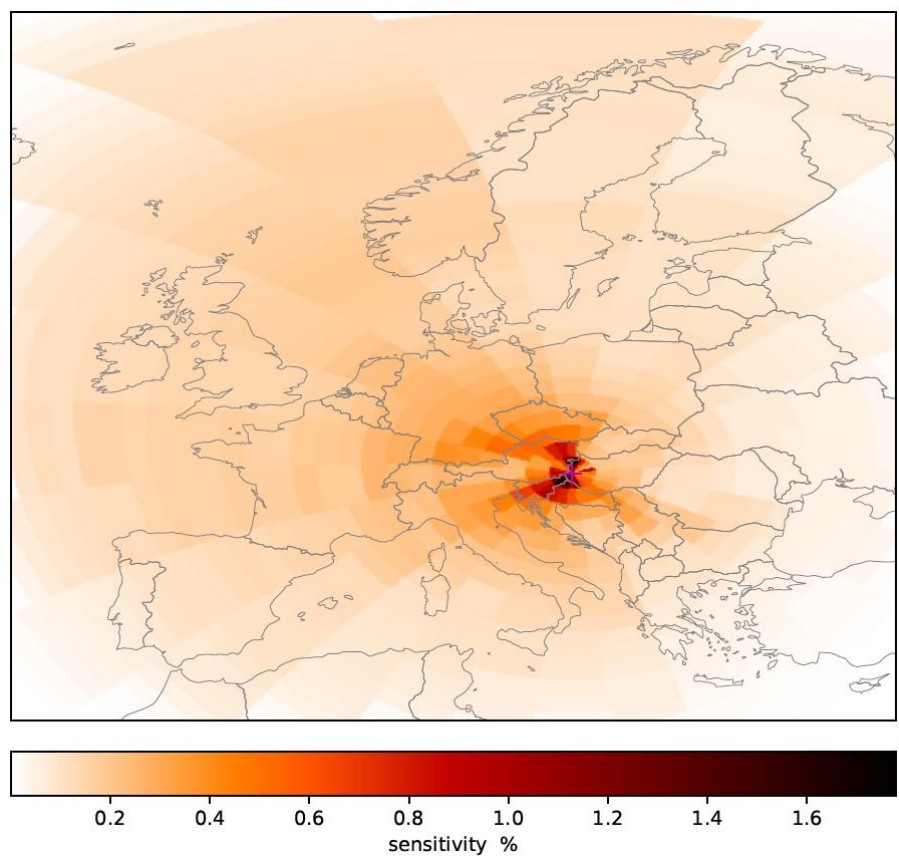


**Figure 7: Footprint climatology of the concentration measurements performed at 115 m elevation above the ground in 2019 calculated by the STILT model (Carbon Portal, 2022).**

The 'hot spot' event mentioned above occurred between 11 and 12 o'clock local time on 4 February when the
light wind ($0.8$ m s$^{-1}$) from the center of the village (wind direction: 300°) directly carried pollution to the measurement sensor in an extremely unstable atmosphere (Pasquill-Gifford stability class A). This process increased the background concentration by 13.6 nmol mol$^{-1}$. This single-hour measurement has to be highlighted in the quality control process and flagged as a regionally non-representative, locally influenced event.

Carbon dioxide and nitrous oxide behave similarly to carbon monoxide during the short transport time from the
village to the tower. Consequently, their excess concentrations caused by the emission in the village are proportional to their emission densities relative to those of carbon monoxide, and the shape of the frequency distributions is the same (Fig. 8). In the case of carbon dioxide, there were only two events (hourly data points) when the excess concentration exceeded 0.04 µmol mol$^{-1}$ (0.06 µmol mol$^{-1}$, and 0.18 µmol mol$^{-1}$ in the extreme case discussed above). The values are within the uncertainty of the measurements. Emissions of nitrous oxide
were relatively low causing only a maximum of 0.03 nmol mol$^{-1}$ excess (the extreme value discussed above is 0.10 nmol mol$^{-1}$), which was practically undetectable.

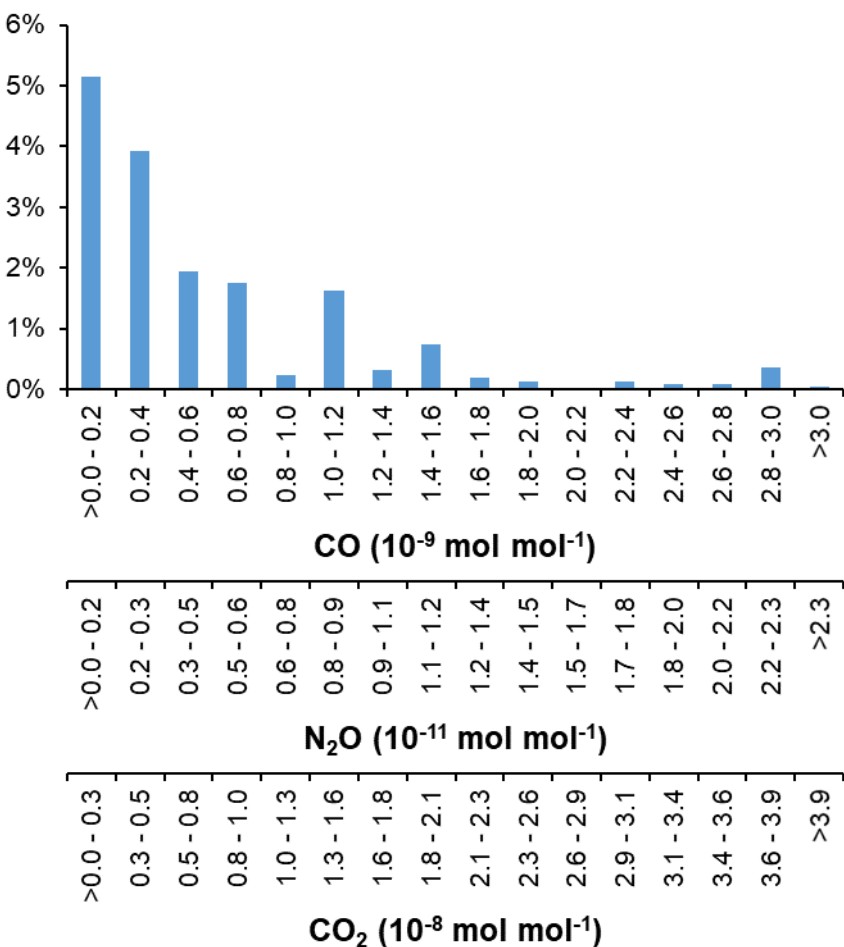

**Figure 8: Frequency distribution of the excess concentrations at the measuring point allocated to emission sources in the village. Cases of zero excess (83.2 %) are not presented. The distribution is the same for all components but the scales are different. Note: scales for $N_2O$ and $CO_2$ are rounded and based on the CO scale.**

The wintertime (Dec-Feb) average excess of CO, $N_2O$, and $CO_2$ concentrations are 0.10, <0.01, and 1.34 nmol mol$^{-1}$, respectively. Due to the lack of industrial and commercial activities, residential heating is the dominant emission source in the village. This means that the excess concentrations of the village may be even lower in the non-winter seasons. The low values confirm the regional representativeness of the measured concentration data, i.e. that the Hegyhátsál tower qualifies as a regional background monitoring site.

**4 Summary and conclusion**

In this study, we have shown that tall-tower eddy covariance measurements may be used for the determination of the emission of a smaller region even if it occupies only a minor portion of the footprint area of the measurements. The study is presumably the first one aiming at the direct measurement of GHG emission of a small rural settlement, while similar measurements have already been performed in urban environments that typically exhibit different emission characteristics. Although the results are subject of significant uncertainties mainly due to the low number of available data points, the results reveal that while the emission factor-based calculations of carbon monoxide and carbon dioxide emissions do not differ significantly from the real-world

top-down measurements, the emission factor-based nitrous oxide emission is significantly underestimated. The difference is remarkable and needs explanation even if we consider the uncertainties of both the bottom-up and the top-down calculations presented here. Further in-depth studies are needed, which could potentially result in a correction of the emission factors. The relatively high CO to $CO_2$ ratio and the high $N_2O$ emission density suggest a higher than "officially" assumed contribution of biomass burning, and a possibility of illegal waste burning.

Using a 3D transport model, we confirmed that the village, as a local pollution source, hardly influences the concentration measurements at the nearby greenhouse gas monitoring station at 82 m height. Hence the site can be qualified as a regional background monitoring site.

*Data and code availability.* The raw data of the calculations are available from the corresponding author. The FFP footprint model code is publicly available at https://footprint.kljun.net/, while the GRAL model code can be downloaded from the Technische Universität Graz (https://gral.tugraz.at/index.php/download).

*Author contributions.* Conceptualization, part of the calculations, writing the paper (L.H.); flux calculations (Z.B.); forward transport modeling (Z.F.); statistical background calculations (R.H.); landscape information and processing (A.K.); footprint modeling support and editing the paper (N.K.). All authors have read and agreed to the published version of the manuscript.

*Competing interests.* The authors declare that they have no conflict of interest.

*Acknowledgments.* The authors highly appreciate the information on the official emission calculations and emission data received from Gábor Kis-Kovács (Unit of National Emission Inventories, Hungarian Meteorological Service).

*Financial support.* This research was funded by the Hungarian National Research, Development and Innovation Office (grant no. OTKA K129118/K141839 and FK128709), by the Széchenyi 2020 programme, the European Regional Development Fund and the Hungarian Government (GINOP-2.3.2-15-2016-00028), and also by the RRF-2.3.1-21-2022-00014 project (National Multidisciplinary Laboratory for Climate Change). A. Kern was supported by the János Bolyai Research Scholarship of the Hungarian Academy of Sciences (grant no. BO/00254/20/10). Z.B. and A.K. were supported by grant "Advanced research supporting the forestry and wood-processing sector's adaptation to global change and the 4[th] industrial revolution", No. CZ.02.1.01/0.0/0.0/16_019/0000803 financed by OP RDE. This work was also supported by the GINOP-2.3.2-15-2016-00055 project, financed by the Ministry of Finance, Hungary.

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
