# Peer review of "Real-world wintertime CO, $N_2O$ and $CO_2$ emissions of a Central European village"

_Atmospheric Measurement Techniques, 2022_

## Author Comment (AC1)

**Authors' response to Referee#1**

First of all, we would like to thank the Referee for his/her positive evaluation of our manuscript and would like to thank him/her for the comments and suggestions, which have helped us to improve the manuscript. Our study was intended to shed some light on a hardly studied question (emission of small settlements) offering a possible methodological solution. Although raw data series was available for six winter seasons, the number of hourly data useable for the estimation of the emission from the village was low (only 44-64 data points depending on the criteria used to filter the data). This low number practically prevented the detailed, quantitative statistical analyses of the results and the scientifically solid uncertainty calculations rightly required by the Referees. Nevertheless, in the revised version of the manuscript, we do our best to present and discuss the uncertainties as much as possible. We add a completely new Supplementary material to present uncertainty-related results that do not fit directly into the main manuscript but seem important to support our assumptions. Below you will find our detailed, point-by-point responses to the comments and suggestions. Responses are given in **blue**.

Referee's main concerns regarding the methodology and the subsequent results:

1. My initial concern relates to the validity of the estimated long-term (top-down) village emissions, which are then compared to the bottom-up estimations and subsequently used to estimate the influence on the concentration measurements. The Authors use the median values from 44 – 64 hourly estimates (across several years) to estimate indicative 3-month period emission totals (Lines 280 – 282). The used sample size is extremely small compared to the aggregated period (2 – 3 %) and it is thus very difficult to extract safe conclusions for a parameter with such intense hourly variability as the household emissions.

The variability of household emissions would follow a diurnal pattern according to local population habits for heating (and cooking) along a temporally aggregated pattern (detectable from daily to monthly steps) which would follow mainly temperature. Hence, the sample used can be strongly biased according to the above patterns, especially when considering the multiple climate-related filters on the observation data, which would favour daytime and clear-weather observations.

The interquartile ranges (Table 1) confirm the high variability of village emissions and the related uncertainties in the approach. I suggest the Authors to perform an analysis on the available dataset to examine the possible biases that could affect the conclusions (e.g. statistics on hour of day, type of day, temperature compared to Dec-Feb period, etc.). Furthermore, to present the results with caution, providing uncertainty ranges where possible, and discussing the data representativity, potential biases and their effects on the conclusions.

Greenhouse gas monitoring stations are either located in an urban environment for the monitoring of the urban emissions or at locations where the anthropogenic disturbance is the minimum. Neither of them is really suitable for the monitoring of the understudied small settlement emission. In our essentially methodological study, we use a "background" monitoring site located in the vicinity of a small village, which is expected to provide at least a limited amount of data on the emission of a village. The location of the monitoring station was selected to be as free from direct anthropogenic disturbance as possible, which inevitably

leads to the low number of usable data for our present study. However, we think that the novelty of our approach and the information provided are interesting for the scientific community despite the unavoidable uncertainties.

Fig. 5 in the original manuscript presents how quickly the number of available hourly data decreases with the increase of the prescribed minimum footprint coverage of the village (integral of the footprint function over the village, alpha in Eq. (1)). In the new Supplementary material, figures will show how $F_{village}$ converges to the assumed true value with the increasing alpha minimum, while the interquartile range also decreases indicating a less and less uncertain emission value. An example figure is presented here (Fig. 1) for $N_2O$.

[Figure]

**Fig. 1: Convergence of the median emission density (orange line) and interquartile range (grey bands) as the function of the minimum footprint coverage of the village ($\alpha_{min}$). The vertical yellow lines indicate the values presented in Table 1 in the main text of the paper ($\alpha \geq 0.25$ and $\alpha \geq 0.30$). The number of data available for the calculations is shown in Fig. 5 in the main text.**

In addition to the fuel used, the emission of a household depends on the outside temperature and the time of day. The vast majority of the calculated $F_{village}$ values relate to the period of 9-15 h local standard time (57/64 in the case of $\alpha \geq 0.25$, 42/44 in the case of $\alpha \geq 0.30$). Although there are no quantitative data on the diurnal heating/cooking behavior of the populations, it can be assumed that our data avoid both the nighttime low emission period and the morning/evening high emission periods, and hence should be representative of the diurnal average emission values. This hypothesis cannot be tested with the available data, and the resulted uncertainty must be taken into account in the interpretation of the results. This reasoning is inserted into Section 3.2 (Emission from the village) of the revised manuscript.

The heating intensity obviously depends on the outside temperature. Emission of a city, region, or country is usually given for the average environmental conditions, therefore, the study also provides emissions from the village for average environmental winter conditions. The low number of data not making it possible to demonstrate a reliable temperature-dependence is another reason for avoiding the discussion of the temperature-dependence. The emission data reported in our manuscript (Table 2 in the original manuscript) are totals for the whole winter season. However, in response to the request of the Referee, the new Supplementary material will present the indicative temperature dependence of the emissions,

as much as the low number of data makes it possible. The figure below (Fig. 2) is one of the three new graphs that will be included in the Supplementary material.

[Figure]

**Fig. 2: Temperature dependence of the CO emission densities in the village for α≥0.25 and α≥0.30. Medians in each temperature range, and the medians of all available data (blue and orange lines). No median is calculated for n<9.**

All these sources of uncertainties, the assumptions and the procedures followed will be discussed in the revised 'Results and discussion' section of the paper.

2. The second main concern is about the assumption of homogeneity in the "natural fluxes". By keeping Fnatural constant on Eq. 2, temperature variability would induce some bias on the estimation of village emissions. Moreover, directionality could be significant and affect the medians used as representative natural fluxes. I wonder if the conclusion regarding the extremely high N2O emissions is somehow affected by the cropland emissions in the direction of the village. Furthermore, the major road on the south and the west of the station could be a confounding factor on the natural flux estimation (mainly concerning the southern wind directions). I suggest the Authors to provide detailed analyses to examine the validity of temporal and directional homogeneity hypothesis and the potential errors associated with it, as well as to discuss the main effects on the methodology and the results.

Our method assumes homogeneous and isotropic spatial distribution of GHG fluxes from the "natural" landscape. To evaluate whether this condition was met, the measured flux values were grouped into wind sectors of 22.5-degrees, and the median value was calculated for each sector. (See the example figure below (Fig. 3). Figures for all the three gases studied will be presented in the new Supplementary material.) The directional medians were compared with the overall median of the data set using the asymptotic K-sample Brown-Mood median test. Only the median of the southwest direction deviated significantly ($p<0.05$) from the overall median, possibly due to a flux signal from the major road in that direction. Discrimination of a specific direction would have been inconsistent with the otherwise applied α-based filtering, therefore we calculated how much the somewhat higher fluxes from this direction contributed to the overall median. Rejection of the data from the southwest direction would reduce the median CO emission density from 139 ng m$^{-2}$ s$^{-1}$ to 128 ng m$^{-2}$ s$^{-1}$. The corresponding values

[Figure]

**Fig. 3: Sectorial distribution of the median fluxes of $N_2O$ (ng m$^{-2}$ s$^{-1}$) of the „natural" landscape (red), and the median flux calculated using all available data (black). No median is calculated for n<10. A logarithmic scale is applied for better visualization. The village is located in the SW-W-NW-N sector (see Fig. 3 in the main text of the paper). Similar figures are presented for CO and $CO_2$ in the new Supplementary material.**

for $N_2O$ are 5.9 ng m$^{-2}$ s$^{-1}$ and 5.8 ng m$^{-2}$ s$^{-1}$, while for $CO_2$ 12 µg m$^{-2}$ s$^{-1}$ and 11 µg m$^{-2}$ s$^{-1}$, respectively. These small deviations hardly influence the calculation of the emissions from the village and are negligible relative to the calculated emissions from the village. Therefore, for keeping the consistency of the filtering method, all available data were used for the calculation of the median emission densities of the "natural" landscape. The sector containing the village could not be tested because of the contribution of the village itself. We hence assumed that the overall median of the "natural" GHG fluxes was also applicable for this sector.

$F_{natural}$ shows a moderate positive temperature dependence for all gases studied. As we focused on the average wintertime environmental conditions we used the median $F_{natural}$ calculated from all available data. Nevertheless, the Supplementary material will include figures on the temperature dependence of $F_{natural}$ because the information might be useful for other studies.

All these sources of uncertainties, the assumptions, and the procedures followed will be discussed in the revised 'Results and discussion' section of the paper.

3. The third concern goes back to the main question raised by the Associate Editor Prof. Domink Brunner. In my first reading of the article I was left with the impression that Eq. 2 was not finally used in the estimation of village emissions and the weighting factor alpha (α) was only used to filter which Eddy Covariance measurements are affected significantly by the village emissions (Fig. 5). But if this was the case, then the subsequent calculations of total village emissions (lines 280 – 282) would not be valid, so I assumed that the method described in section 2.1 was applied. To avoid confusion, I ask the Authors to further clarify the method used to estimate the village emissions. There is a number of unclear points:

- Statistics from two Fvillage estimates are given, at a = 0.25 and a= 0.3. Does this mean that the indicated value is the lower threshold used to apply Eq. 2?

We apologize for the obscure description of the method. Obviously, for the calculations, Eq. (2) was applied. It will be clearly stated and explained in the revised version of the manuscript. The emission values are given for $\alpha \geq 0.25$ and $\alpha \geq 0.30$. The cause of the obscurity was likely the faulty use of the = sign instead of $\geq$, which is corrected in the revised manuscript and the revised header of Table 1.

- Is Eq. 2 applied at the hourly Fmeasured and α values, keeping Fnatural constant, to derive the Table 1 statistics for Fvillage?

Yes, it is. It will be more clearly stated in the revised manuscript.

"The emission density of the village has been calculated for $\geq 25$ %, and $\geq 30$ % footprint weighted coverage ($\alpha \geq 0.25$ and $\alpha \geq 0.30$) using the constant $F_{natural}$ and the actual hourly α values in Eq. (2)."

- The description of the concept in Lines 85 – 96 is not totally clear, confounding flux densities (Fx) with their theoretical areal attribution, without specifically defining the equation of α which is the tool used for this attribution. I think this part needs to be more detailed, giving the Equation used for estimating α.

α is the integral of the footprint function for the territory of the village. The detailed mathematical description of the footprint function can be found in Kljun et al. (2015). We do hope that the rephrased description will be more understandable for the reader than before:

"The measured flux ($F_{measured}$) is the combination of the fluxes originating from the built-up areas ($F_{village}$) including the houses, farm buildings, backyards, roads, and parks within the village, and those of the non-residential areas, which we denote here as "natural" landscape ($F_{natural}$):

$$F_{measured} = \alpha * F_{village} + (1 - \alpha) * F_{natural} , \qquad (1)$$

where α is the contribution of the village within the footprint area. Note that the contribution of the surface flux to the measured flux is not uniform within the footprint area (Schmid, 1994; Kljun et al., 2002; Vesala et al., 2008). The weighted contribution of a surface source at a specific unit area to the measurement at the tower at each time step can be estimated by using the footprint function value at the given point. The integral of the footprint function over the infinite x-y plane equals one. Hence, with a suitable footprint model, if $F_{natural}$ is known, then $F_{village}$, the emission density of the village within the footprint area can be calculated as follows:

$$F_{village} = (F_{measured} - (1 - \alpha) * F_{natural}) / \alpha \qquad (2)"$$

Specific comments:

Lines 43 – 45: An appropriate reference regarding household emissions is missing.

It is an everyday experience that in small settlements the heating-cooking appliances used depends on the local socio-economic conditions, cultural traditions, and available infrastructures. We do not know about any sociological survey in Hungary/Central Europe quantifying the differences among the villages under different conditions, different regions, and differences from the more uniform cities. There are numerous publications about the emissions of households in cities, which may be quite different from those in small settlements. Data on village emissions are available from China, India, and Africa but these are hardly relevant for a European village. The few publications available for Europe mainly focus on particulate matter emission because it is critical for the health of the local inhabitants.

Lines 57 – 58: A reference regarding the uncertainties related to residential heating in the emission inventories is missing.

We agree with the Referee that this information would be useful but according to our best knowledge, such information is not available in either the literature or the national greenhouse gas inventories. There are emission factors and related uncertainties for well-defined heating methods (e.g. natural gas, LPG, lignite, coal, etc.). However, in a village, especially in less developed conditions, people use different fuels (e.g. natural gas, coal, wood, waste, and anything burnable) in unknown ratios. There is no survey and quantitative data on it, which implies a significant uncertainty in any theoretically calculated emission. Realistic values could only be obtained by performing as many measurements as possible. Our paper is an attempt to provide some information even if the uncertainty of the resulted data is presumably high and cannot be clearly defined.

Lines 85 – 96, 214 – 215: The description of the method is not very clear (see main point no. 3). It is important to clarify the equation used for estimating α.

The sections will be reformulated in the revised version of the manuscript. See our response above.

Line 111: Please provide the orientation of the sonic anemometer. Are there wind flow disturbances from the tower structure at some wind directions?

The sonic anemometer is mounted on an instrument arm projecting to the north. A certain disturbance can be revealed when the wind is from the south. We correct this effect as it is described in Barcza et al. (Agricultural and Forest Meteorology 149, 795-807, 2009). The following sentences will be inserted into the revised manuscript: "*The eddy covariance system is mounted on the tower at 82 m above the ground, on an instrument arm of 4.4 m long*

*projecting to the north. The disturbance of the flow pattern during southerly winds is corrected as described in Barcza et al. (2009).*"

Lines 135 – 137: Is this the only data quality check applied on the EC fluxes? What about other quality flags such as steady state and integral turbulence characteristics tests, technical failures, weather effects?

Only quality-checked data were involved in this study. Among these data, those were filtered out when the height of the planetary boundary layer was less than 100 m. (Note: the measurement elevation is 82 m.) The complete quality check procedure can be found in Haszpra et al., (2005) and Barcza et al. (2020) referred to in the original manuscript. The modified text goes as: "*Taking into account the elevation of the EC system of 82 m above the ground, we removed all flux values from the quality-checked data series (Barcza et al., 2020; Haszpra, et al., 2005; 2018) when the top of the boundary layer was below 100 m*".

Lines 222 – 224: Please clarify the input dataset concerning the village emissions and the temporal resolution and the time period of the simulations. Is the mean estimated Fvillage used for hourly simulations?

For the estimation of the effect of the emission in the village on the concentration measurements at the tower, the emission densities calculated in this study were used ($F_{village}$). The model was run in hourly resolution for the period from December 2017 to February 2018. These details will be added to the revised manuscript. Section 2.6 is extended by the following sentence: "*The simulation was run at 10 m horizontal and 3 m vertical resolution for the period of December 2017 – February 2018. at 1 h temporal resolution.*" A further sentence is inserted into Section 3.3: "*For the calculations, $F_{village}$ presented in Table 1 for α≥0.3 was assumed, with a homogeneous distribution over the area of the village.*"

Lines 227 – 244: This part in more related to the theory or the basic concept descriptions, rather than the results section.

The methodological section is about the basic concept, the model, the measurement conditions, and the input data. Lines 227-244 are an introduction to the calculation of the "natural" emission. This part gives the origins of greenhouse gases in the unpolluted atmosphere, and a few words about the factors influencing their emissions. If we moved this section to the methodological chapter, it would be too far from the actual use of the information. We hope that the Referee will accept keeping this part in its original place.

Lines 267 – 270: The method used for the estimation of village emissions is not clear (see main point no. 3).

The cause of the obscurity of the sentence is the faulty use of the = sign instead of ≥. The sentence is corrected: "*The emission density of the village has been calculated for ≥25 %, and ≥30 % footprint weighted coverage (α≥=0.25 and α≥=0.30) (Fig. 6). In addition to the median flux, Table 1 gives the estimated lower and upper quartiles, and the number of footprints available for the calculations.*" The header of Table 1 is also corrected accordingly.

Lines 300 – 332: The description of the bottom-up emission estimation approach would be better related to the methodology section rather than the results.

Section 3.2 is about the emission from the village where our top-down calculations are compared with the bottom-up estimations. Therefore, it seems reasonable to keep all data together to save the reader from scrolling back and forth between chapters. We hope that the Referee can accept this argument.

Lines 334 – 335: This statement is arbitrary. Uncertainty ranges are not given to enable comparison and similarity conclusions between the datasets.

The Referee is right, there are no uncertainty estimates available for either the bottom-up estimation or the top-down estimation. The sentence is reformulated as follows: "*Although these statistics-based, bottom-up numbers seem to underestimate the CO and CO$_2$ emissions calculated by the top-down approach, - when taking into account the rough estimate of the bottom-up emission and the uncertainties of the top-down approach – the results are similar enough to support the applicability of our method..*"

Lines 401 – 402: This statement is a bit strong and not entirely supported considering the uncertainties related to the results.

The sentence is reformulated as follows: "*In this study, we have shown that tall-tower eddy covariance measurements may be used for the determination of the emission of a smaller region even if it occupies only a minor portion of the footprint area of the measurements.*"

Section 3: The discussion of the results is absent. The Authors should develop this part of the manuscript considerably, given that the methodology and the results raise several and substantial issues and questions which should be adequately covered (see also main points 1 and 2).

The section will be significantly expanded including a detailed description of the limitations of the study. We will discuss the questions of the directional homogeneity and temperature dependence of the "natural" emission, the minimum village coverage required, the temporal variation of the emissions, and the arguments for the applied procedures.

Finally, two points for potential consideration and discussion:

- It is evident that a considerable area covered by croplands, gardens and other green areas is classified as urban (Lines 161 – 165, Fig. 3). This misattribution of land cover to emissions which is not functionally associated to could have an impact on the spatial weighting approach used to estimate urban emissions (α factor in Eq. 1 and 2). It would be interesting to examine the effects of this theoretical attribution on the final results by restricting the definition of the village territory to the "actual" built-up area compared to the existing definition.

The actual built-up area is only 8.5 % of the village's total area. For the smaller area, a higher emission density will be received. The total emission of the village is the emission density multiplied by the area of the emission, and thus – theoretically – the total emission discussed in the paper may not change. In practice, the uncertainty may increase significantly because the stationarity requirement of the footprint model providing the hourly average footprint functions may be fulfilled less at higher spatial resolution. Except for the large point/stationary sources, the anthropogenic emissions are always given as area sources for administratively defined territories.

- CO appears to be the trace gas that is most efficiently discriminated between village and background values (Table 1). Past studies have used it as an indicator of fossil fuel CO2 emissions by employing some standard CO:CO2 ratio. Having very little experience on the subject, I wonder if the applied spatial weighting methodology can be complementary to a methodology that estimates village emissions by using CO as an indicator.

$CO:CO_2$ ratio is fuel-specific and may also depend on the condition of the heating/burning appliance, chimney, etc. In the village, people may use natural gas, coal, brown coal, lignite, wood, etc. – all with different $CO:CO_2$ ratios. Radiocarbon measurements suggest that the ratio of fuel containing "modern" carbon (wood, agricultural waste, etc.) relative to the fuel containing fossil carbon (natural gas, coal, lignite) may be high in the region (Major et al., Temporal variation of atmospheric fossil and modern $CO_2$ excess at a Central European rural tower station between 2008 and 2014. Radiocarbon 60, 1285-1299, 2018). Unfortunately, the temporal resolution of the radiocarbon measurements does not allow the precise allocation of the sources, and the footprint of the measurements is also quite different from the spatial scale of the present study.

---

## Author Comment (AC2)

**Authors' response to Referee#2**

First of all, we would like to thank the Referee for his/her positive evaluation of our manuscript and would like to thank him/her for the comments and suggestions, which have helped us to improve the manuscript. The point-by-point responses given to the comments and suggestions are printed in **blue**.

The paper proposes an original analysis of CO, CO2 and N2O eddy flux measurements made from a tall tower, in order to identify surface emissions at the scale of a village. The presentation of the measurement site, the eddy-fluxes and concentration measurements as well the tools used for the footprint calculations are short but rather clear. The approach is really interesting, and represents an interesting valorization of flux measurements. Very clearly, the main question concerns the uncertainties in the emission estimates, both by the bottom-up method and derived from the eddy flux measurements. Consequently, the main general comment would be to give as much as possible an estimate of the ranges of possible values considering the main identified uncertainties.

Our study was intended to shed some light on a hardly studied question (emission of small settlements) offering a possible methodological solution. Although raw data series was available for six winter seasons, the number of hourly data useable for the estimation of the emission from the village was low (only 44-64 data points depending on the criteria used to filter the data). This low number practically prevented the detailed, quantitative statistical analyses of the results and the scientifically solid uncertainty calculations rightly required by the Referees. Nevertheless, in the revised version of the manuscript, we do our best to present and discuss the uncertainties as much as possible. We add a completely new Supplementary material to present uncertainty-related results that do not fit directly into the main manuscript but seem important to support our assumptions. Below you will find our detailed, point-by-point response to the comments and suggestions.

Few specific comments:

Introduction: could you provide an estimate of the percentage of CO2 emissions related to villages less than few thousands inhabitants either at the scale of Hungary or Europe ?

Unfortunately, we cannot. According to the recent IPCC Report urban areas are associated with approximately two-thirds of the total anthropogenic $CO_2$ emissions, however, the definition of "urban area" may be different in different publications. In the case of most countries, urban-scale emission data are available only for the (mega)cities. Emission data for small settlements are not available for Hungary and the neighboring countries under similar conditions, at least according to our best knowledge.

"we removed all flux values for the periods when the top of the boundary layer was below 100 m": Given that you have vertical profile measurements of trace gases and meteorological data, wouldn't it be possible to derive the periods with low boundary layer height from observations rather than from ERA5 ?

In an MSc thesis, we tried to use the tall-tower vertical profile measurements for the determination of the height of the planetary boundary layer (PBL). (Unfortunately, the thesis is available only in Hungarian.) Due to methodological issues, the maximum stable boundary layer height that could be reliably derived from the vertical profile data was only 65 m. The eddy covariance system used in this study is mounted above this elevation, at 82 m. Identification of the cases, when the PBL height was at least 100 m, was only possible using the ERA5 database. The flux measurements may be questionable in very stable conditions, like when the top of the PBL is close to the measurement elevation or below that.

"The emission densities of CO, N2O, and CO2 obtained for the natural landscape are 139 ng m-2 s-1, 5.9 ng m-2 s-1, and 12 μg m-2 s-1, respectively": according to the criteria used for the PBL development I guess your methods favors daytime events, compared to nighttime. Still do you have cases for the night ? If so do you see a difference between day and nighttime fluxes ? I do not expect strong diurnal variability for the natural fluxes in wintertime, but for the anthropogenic emissions it must very significant.

Concerning the "natural" emission, there are more daytime data than nighttime ones, however, the difference is not big as it will be presented in the new Supplementary material added to the main paper, and can be seen here in Fig. 1. The daytime-nighttime emission differences are not negligible but do not seem significant at the general uncertainty of the flux data (Fig. 2 below, figures for all three gases will be available in the Supplementary material). For climatic reasons (specific wind direction occurring under specific weather conditions), emission data for the village are available only during daytime hours (Fig. 3). Although there are no quantitative data on the diurnal heating/cooking behavior of the populations, it can be assumed that our data avoid both the nighttime low emission period and the morning/evening high emission periods, and hence should be representative of the diurnal average emission values. This hypothesis cannot be tested with the available data, and the resulted uncertainty must be taken into account in the interpretation of the results. This reasoning is inserted into Section 3.2 (Emission from the village) of the revised manuscript.

[Figure]

**Fig. 1: Temporal distribution of the number of hourly flux data available for the determination of the emission density of the „natural" landscape**

[Figure]

**Fig. 2: Temporal distribution of the median emission density of the „natural" landscape, and the calculated overall median emission density (horizontal red line) for $N_2O$.**

[Figure]

**Fig. 3: Temporal distribution of the number of hourly emission density data available for the village at ≥25 % and ≥30 % footprint coverage**

"Our concept assumes that both the "natural" landscape … are homogeneous from the point of view of emission density": would it be possible to elaborate on this assumption especially for CO2 ?

Our method assumes homogeneous and isotropic spatial distribution of GHG fluxes from the "natural" landscape. To evaluate whether this condition was met, the measured flux values were grouped into wind sectors of 22.5-degrees, and the median value was calculated for each sector. (See the example figure below (Fig. 4). Figures for all the three gases studied will be presented in the new Supplementary material.) The directional medians were compared with the overall median of the data set using the asymptotic K-sample Brown-Mood median test. Only the median of the southwest direction deviated significantly ($p<0.05$) from the overall median, possibly due to a flux signal from the major road in that direction. Discrimination of a specific direction would have been inconsistent with the otherwise applied α-based filtering, therefore we calculated how much the somewhat higher fluxes from this direction contributed to the overall median. Rejection of the data from the southwest direction would reduce the median CO emission density from 139 ng m$^{-2}$ s$^{-1}$ to 128 ng m$^{-2}$ s$^{-1}$. The corresponding values for $N_2O$ are 5.9 ng m$^{-2}$ s$^{-1}$ and 5.8 ng m$^{-2}$ s$^{-1}$, while for $CO_2$ 12 μg m$^{-2}$ s$^{-1}$ and 11 μg m$^{-2}$ s$^{-1}$, respectively. These small deviations hardly influence the calculation of the emission from the village and are negligible relative to the calculated emissions from the village. Therefore, for keeping the consistency of the filtering method, all available data were used for the calculation of the median emission densities of the "natural" landscape. The sector containing the village could not be tested because of the contribution of the village itself. We hence assumed that the overall median of the "natural" GHG fluxes was also applicable for this sector.

[Figure]

Fig. 4: Directional distribution of the median CO2 flux (μg m$^{-2}$ s$^{-1}$) of the „natural" landscape (red) and the median flux calculated using all available data (black). There is no data from the direction of the village. Similar figures will be presented in the Supplementary material for $N_2O$ and CO. Logarithmic scale is applied for the better visualisation.

Table 2: The estimation of the emissions by the so-called bottom-up method is well explained in the text, but given the assumptions (at some point in the text an uncertainty of a factor of 2 is considered) , it would be preferable to give the range of possible emissions â    â    rather than a precise value.

The bottom-up estimations given in Table 2 are the results of several weakly supported assumptions, therefore, only the two extreme cases (everybody uses only natural gas where available/nobody uses natural gas at all) could define the ranges. We do not think that these non-realistic extreme values would orient the readers concerning the uncertainty of the data. Instead, the uncertainties are discussed in the text

Concerning the N2O, it would be interesting to develop the discussion to develop if it is realistic to have an order of magnitude of difference because of the uncertainties on the burning of waste?

According to our best knowledge, waste burning studies focus only on particulate matter and organics emissions because of their health risk. We do not know about any study measuring or estimating $N_2O$ emission from household waste burning, therefore, we cannot answer the question of whether waste burning explains the high $N_2O$ emission. Nevertheless, our previous study, Haszpra et al. (2019) referred to in the original manuscript and using a different methodology, also indicated that the $N_2O$ emission may be much higher than what would be expected from the activity data and the used emission factors. The magnitude difference between the calculated bottom-up and top-down emissions seems too high to think it is only an uncertainty issue of the methods. Taking into account that nitrous oxide is a potent greenhouse gas, studies should be initiated to identify the unknown or underestimated source(s).

The section on the representativeness of the tall tower as regard of atmospheric mole fractions measurements appears decoupled from the main discussion of the paper, but still this is an interesting contribution, but it might make more sense to have this discussion first in the paper.

The Hegyhátsál tall-tower GHG monitoring station was established as a regional background monitoring site, as free from direct anthropogenic influence as it is possible in the highly industrialized, densely populated Central Europe. The selection of the location was based on expert considerations. Later, the diurnal variation of SF6 concentration, the lack of accumulation in the nighttime boundary layer, indicated that the human influence at the site is low indeed. In the present study, the site is explicitly used for the estimation of anthropogenic emissions, which seems to contradict the "background" status of the monitoring site. For resolving this contradiction it seemed necessary to discuss the significantly different footprints of the flux and concentration measurements. It seemed logical to go into these details only after showing up that the site can give information on the anthropogenic emission. We think if this topic were discussed at the beginning of the paper the readers would not understand why this topic is important at all.

---

## Referee Report (RR1)

Review of the manuscript: " Real-world wintertime CO, N2O and CO2 emissions of a Central European village" by László Haszpra et al. (Atmos. Meas. Tech. Discuss. , https://doi.org/10.5194/amt-2022-39)

The authors made a real effort to answer the questions posed during the first review. The limitations of the results are now better explained, and the supplementary information is relevant.

---

## Author Response (AR2)

Dear Editor,

Following Reviewer#2's suggestion, we performed additional calculations. We have added an extra paragraph to the end of Section 3.2 about the results of these calculations. The results do not change the previously reported data and the conclusion of the paper, therefore we have not made any more significant changes in the text of the manuscript. An extra table (Table S1) has been added to the Supplement containing the results of the additional calculations.

Yours sincerely

László Haszpra

Response to Reviewer#2

We thank the Reviewer for his/her suggestion. Following the suggestion, we recalculated the emission density of the village taking into account the likely diurnal and temperature variations in the natural emissions ($F_{nature}$). A simple bivariate regression model could not be applied because of the high non-Gaussian scatter of the low number of data and the cross-correlation between time and temperature. Instead, we grouped the natural fluxes into 3-hour time windows (0–3 h, 3–6 h, 6–9 h,... local standard time), and within each time window the data were grouped into 3-degree temperature ranges centrally to 0 °C (-10.5 – -7.5 °C, -7.5 – -4.5 °C,-4.5 – -1.5 °C, -1.5 – +1.5 °C,...). The median flux was calculated for each group including at least 10 data. These median fluxes presented in the new Table S1 in the Supplement were used as $F_{natural}$ for the recalculation of $F_{village}$ taking into account the actual temperature and time of the day. For the 3 cases for which the median was not available, the median of the nearest temperature range was used. The calculated emission densities for CO, $N_2O$, and $CO_2$ from the village at $\alpha \geq 0.3$ are 3.6 µg m$^{-2}$ s$^{-1}$, 36.3 ng m$^{-2}$ s$^{-1}$, and 73 µg m$^{-2}$ s$^{-1}$, respectively, practically the same as using the constant $F_{village}$ (median of the overall dataset). The $N_2O$ emission is lower by approximately 15 % according to this method but it does not question the conclusion, namely, the official $N_2O$ emission is significantly underestimated. We have added an extra paragraph to the end of Section 3.2 (lines 382-392 in the revised manuscript) about the method and the results of the above calculations and left the other part of the manuscript unchanged.

Minor comments:

*Lines 215-216: Please add in parenthesis that to derive α, the integral concerns all grid cells appearing in Fig. 3.*

An extra sentence has been added in lines 216-217 of the revised manuscript.

*Figure 5 Caption: Shouldn't the 4277 be 1120?*

x=0 ($\alpha_{min}$=0) means α≥0, that is all available data (background [α=0, 1120 data] and village of any footprint coverage [α>0]), the total number of which is 4277. See line 207.

---

## Author Response (AR3)

Dear Editor,

Thank you very much for reviewing the revised version of our manuscript entitled „Real-world wintertime CO, $N_2O$ and $CO_2$ emissions of a Central European village". We accepted the suggested corrections, and also inserted the Google Earth copyrights into the figure captions as you requested.

Kind regards

László Haszpra